# Combined activation of MAP kinase pathway and β-catenin signaling cause deep penetrating nevi

Iwei Yeh[1,2], Ursula E. Lang[2], Emeline Durieux[3], Meng Kian Tee[1], Aparna Jorapur[1], A. Hunter Shain[1], Veronique Haddad[4], Daniel Pissaloux[4], Xu Chen[1], Lorenzo Cerroni[5], Robert L. Judson [1], Philip E. LeBoit[1,2], Timothy H. McCalmont[1,2], Boris C. Bastian[1,2] & Arnaud de la Fouchardière [4]

Deep penetrating nevus (DPN) is characterized by enlarged, pigmented melanocytes that extend through the dermis. DPN can be difficult to distinguish from melanoma but rarely displays aggressive biological behavior. Here, we identify a combination of mutations of the β-catenin and mitogen-activated protein kinase pathways as characteristic of DPN. Mutations of the β-catenin pathway change the phenotype of a common nevus with BRAF mutation into that of DPN, with increased pigmentation, cell volume and nuclear cyclin D1 levels. Our results suggest that constitutive β-catenin pathway activation promotes tumorigenesis by overriding dependencies on the microenvironment that constrain proliferation of common nevi. In melanoma that arose from DPN we find additional oncogenic alterations. We identify DPN as an intermediate stage in the step-wise progression from nevus to melanoma. In summary, we delineate specific genetic alterations and their sequential order, information that can assist in the diagnostic classification and grading of these distinctive neoplasms.

[1] Department of Dermatology, University of California, San Francisco 94143 CA, USA. [2] Department of Pathology, University of California, San Francisco 94143 CA, USA. [3] Department of Pathology, Centre Hospitalier Lyon-Sud, Lyon 69310, France. [4] Department of Biopathology, Centre Léon Bérard, Lyon 69008, France. [5] Department of Dermatology, Medical University of Graz, Graz 8036, Austria. Ursula E. Lang, Emeline Durieux and Meng Kian Tee contributed equally to this work. Boris C. Bastian and Arnaud de la Fouchardière jointly supervised this work. Correspondence and requests for materials should be addressed to I.Y. (email: Iwei.Yeh@ucsf.edu)

Common nevi are benign melanocytic neoplasms located superficially in the skin. Nevus cells become smaller and less pigmented as their distance from the surface epithelium increases. This characteristic is termed 'maturation' and is an important diagnostic feature to distinguish benign nevi from melanoma[1, 2]. In 1989 Helwig and colleagues described a neoplasm – deep penetrating nevus (DPN) – as an exception to this rule in which the constituent melanocytes maintain their cell size and pigmentation throughout the dermis[3]. Before their initial description, these microscopic characteristcs resulted in the misdiagnosis of almost one-third of DPN as melanoma. While the establishment of DPN as a diagnostic entity has reduced the number of cases falsely classified as melanoma, tumors with the characteristic phenotypic features can present diagnostic challenges, and occasional tumors classified as DPN metastasize with fatal outcome[4].

While the majority of common nevi are clonal proliferations of $BRAF^{V600E}$ mutant melanocytes[5, 6], the genetic drivers of DPN are largely unknown. DPN have overlapping phenotypic features with blue nevi in that they are mainly dermal based and contain numerous melanophages[7]. However, they lack the GNAQ and GNA11 mutations that are found in the majority of blue nevi[8]. DPN also have overlapping phenotypic features with Spitz nevi in that they have large melanocytes with abundant cytoplasm and have previously been shown to harbor HRAS mutations in 6% of cases[8]. To investigate oncogenic alterations in DPN, we sequenced

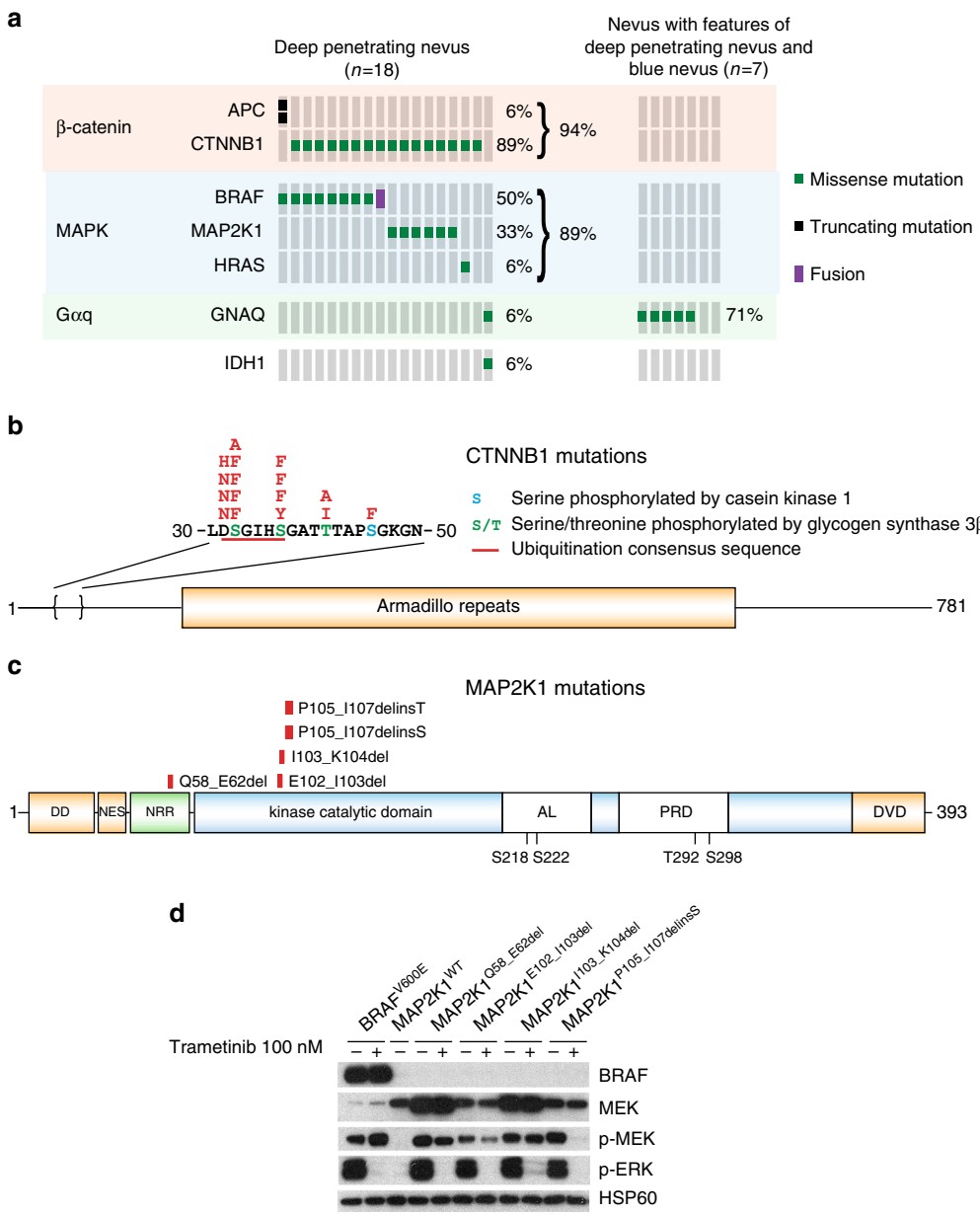

**Fig. 1** Combined MAPK and β-catenin pathway activating mutations define deep penetrating nevi **a** β-catenin pathway mutations, affecting CTNNB1 or APC, and MAPK pathway mutations co-occur in deep penetrating nevi (DPN). Nevi with overlapping features of DPN and blue nevus harbor GNAQ activating mutations and are genetically distinct. BRAF activating mutations are mutually exclusive with *MAP2K1* alterations. **b** *CTNNB1* missense mutations affect codons of a critical domain that is phosphorylated and regulates subsequent ubiquitin-mediated degradation. **c** Indels of *MAP2K1* cluster near a highly conserved lysine within the kinase catalytic domain. One small deletion affects the negative regulatory region (NRR). DD, docking domain for ERK1/2; NES, nuclear export signal; AL, activation loop within the kinase catalytic domain; PRD, proline-rich domain within the kinase catalytic domain; DVD, domain of versatile docking. **d**. MAP2K1 mutations activate MAP kinase signaling, which could be inhibited by the MEK inhibitor trametinib

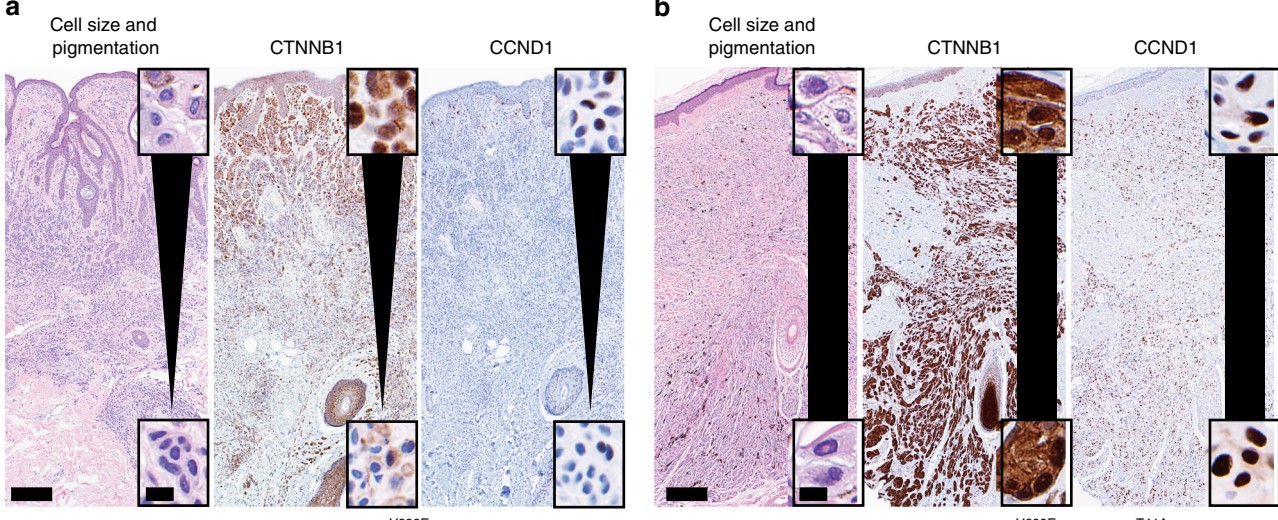

**Fig. 2** Cell size and pigmentation decrease with β-catenin and cyclin D1 levels in common nevi in contrast with deep penetrating nevi. **a** BRAF[V600E] mutant common nevus. *Scale bars*: 300 microns, inset 12.5 microns. **b** BRAF[V600E]/CTNNB1[T41A] mutant DPN. *Scale bars*: 250 microns, inset 12.5 microns. Hematoxylin and eosin staining *left* and immunohistochemistry for β-catenin *center* and cyclin D1 *right*. *Insets* show high power views of melanocytes close to the epidermis *top* and in the deep dermis, away from the epidermis *bottom*. Melanocyte size and pigmentation diminish and β-catenin and cyclin D1 expression levels decrease with distance from epithelium in common nevus **a** but not in DPN **b**

DNA from 18 DPN and 10 common nevi, using a platform that includes exons of several hundred cancer genes and introns involved in oncogenic rearrangements. We sequenced additional nevi with overlapping features of DPN and blue nevi. The resulting sequence data was analyzed for single-nucleotide variants, insertion/deletions (indels), structural variants and copy number changes. We identify MAPK activating mutations in combination with β-catenin activating mutations as a characteristic feature of DPN. We demonstrate that *CTNNB1* mutations arise after *BRAF* mutation in DPN arising from common acquired nevi and confer the characteristic phenotypic features of DPN. In DPN-like melanomas, we identify the presence of additional genetic alterations indicating that DPN are genetically intermediate between benign nevi and DPN-like melanomas.

## Results

**β-catenin and MAPK pathway mutations define DPN.** Cases with typical histological features of DPN were genetically distinct from common nevi and nevi whose features overlapped between DPN and blue nevus. 17 out of 18 (94%) DPN had activating mutations of the β-catenin pathway, mostly affecting *CTNNB1* (16 cases) with rare *APC* mutations (Fig. 1a and Supplementary Table 1), whereas no common nevi harbored β-catenin activating mutations ($P = 0.007214$, binomial test). By contrast, nevi with overlapping features of DPN and blue nevi were devoid of *CTNNB1* and *APC* mutations ($P = 0.03$, binomial test), instead harboring *GNAQ* activating mutations in 5 out of 7 (71%) cases ($P = 0.02$, binomial test). Thus, the latter are better classified as blue nevi based on their genetic profile. Only one DPN harbored *GNAQ*[Q209L] without mutation in *CTNNB1* or *APC*. This tumor also had an *IDH1*[R132C] mutation that was not present in the other *GNAQ* mutant tumors in our series. Perhaps this mutation led to epigenetic changes that altered histopathological features contributing to the apparent morphological misclassification of this lesion that genetically represents a blue nevus.

Most β-catenin pathway activating mutations were missense mutations in exon 3 of *CTNNB1* that disrupt phosphorylation of β-catenin and its ubiquitin-mediated degradation, leading to

increased β-catenin levels (Fig. 1b)[9–11]. In one DPN, two inactivating mutations of *APC* were present, representing an alternative mechanism of β-catenin activation.

16 out of 18 DPN had mutations of the mitogen-activated protein kinase (MAPK) pathway in addition to the activating mutations of the β-catenin pathway, affecting *BRAF*, *MAP2K1* and *HRAS* in a mutually exclusive pattern. 9 out of 18 DPN (50%) had *BRAF* mutations: *BRAF*[V600E] in six cases, *BRAF*[A598_T599insI], *BRAF*[K601E] and *AKAP9-BRAF* fusion in one case each. One DPN harbored an *HRAS*[Q61R] mutation. *MAP2K1* alterations were present in 6 of 18 DPN (33%) consisting of small indels near codons 102 to 107, within helix C of MEK1, which is displaced by the activation loop in the active conformation[12, 13]. In one case, a small deletion affected the negative regulatory region (Fig. 1c)[13]. Similar *MAP2K1* mutations occur in Langerhans cell histiocytosis and hairy cell leukemia, also in a mutually exclusive pattern with *BRAF*[V600E][14, 15].

Three of the five *MAP2K1* indels have been previously observed in melanoma or other cancers, and two have been experimentally demonstrated to constitutively activate MAP2K1[14, 16, 17]. We expressed 4 of the observed *MAP2K1* mutations in 293FT cells and found that they activate MAP kinase signaling, which could be inhibited by the MEK inhibitor trametinib (Fig. 1d). Notably, the *MAP2K1* mutations present in DPN are distinct from those at positions P124 and E203, which often occur in combination with *BRAF* or *NRAS* mutations in melanoma[18, 19].

By contrast, common nevi harbored *BRAF*[V600E] in 7/10, *NRAS*[Q61R] in 3/10, with no additional identified mutations. Thus DPN are genetically defined by the combination of MAPK activating and β-catenin activating mutations and are distinct from common and blue nevi.

To further investigate the role of *CTNNB1* mutation in DPN, we used immunohistochemistry to evaluate β-catenin expression level and localization in DPN and common nevus controls. In common nevi, nuclear β-catenin was expressed in superficial melanocytes situated near the epidermis, whereas the smaller melanocytes in the deeper portions of the nevus had considerably lower expression levels, consistent with prior reports (Fig. 2)[20].

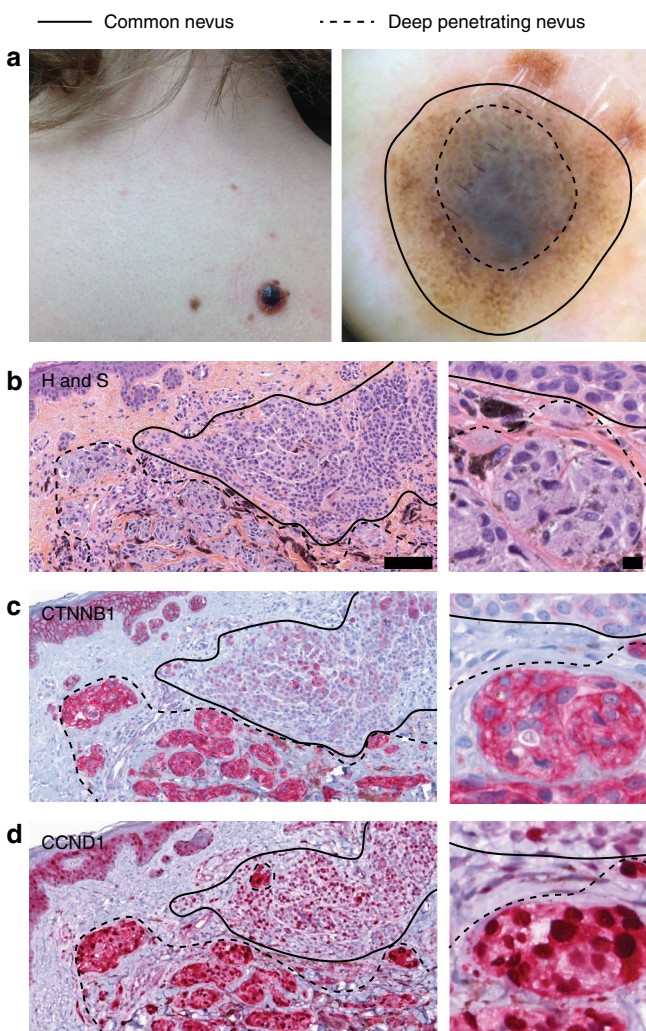

—— Common nevus  ----- Deep penetrating nevus

**a**

**b** H and S

**c** CTNNB1

**d** CCND1

**Fig. 3** Secondary *CTNNB1* mutations in common acquired nevi with BRAF mutations confer the phenotypic characteristics of deep penetrating nevus. **a** Clinical presentation of 'combined DPN' with common nevus and DPN areas. *left* A dark papule grew within a pre-existing nevus on the back of a 31-year-old woman. *right* Dermoscopic image of the tumor. **b** Histopathological features of a 'combined DPN' with common nevus and DPN areas, hematoxylin and safranin (H and S) staining. Melanocytes are enlarged in the DPN portion compared to the common acquired nevi portion. *left* low power view, *scale bar* 100 microns. *right*, high power view, *scale bar*, 10 microns. **c** β-catenin immunohistochemistry. *left* Low power view shows increased cytoplasmic and nuclear β-catenin in DPN melanocytes. In common nevus melanocytes, β-catenin expression is decreased in melanocytes deep within the dermis. *right* High power view. **d** Cyclin D1 immunohistochemistry. *left* Low power view shows increased cytoplasmic and nuclear cyclin D1 expression in the DPN component as compared to the common nevus component

Only deep dermal melanocytes situated near hair follicles also had increased levels of nuclear β-catenin, suggesting that proximity to epithelial structures - epidermis or hair follicle epithelium - may increase β-catenin signaling (Supplementary Fig. 1). In DPN, β-catenin staining was more intense and by contrast did not diminish with distance from epithelia (Fig. 2, Supplementary Fig. 2). We also measured the expression of *AXIN2*, a marker of β-catenin transcription[21], by RNA *in situ* hybridization. In DPN, melanocytes demonstrated constant levels of *AXIN2* expression throughout the neoplasm, whereas in common nevi *AXIN2* expression decreased with distance from epithelium (Supplementary Table 2 and Supplementary Fig. 3).

Cyclin D1 is a direct transcriptional target of β-catenin[22] and is frequently amplified in melanoma[23]. Melanocytes in DPN demonstrated strong and uniform nuclear expression of cyclin D1 by immunohistochemistry. In contrast, immunoreactivity for cyclin D1 in common nevi was limited to melanocytes near epithelia, mimicking the expression pattern of CTNNB1 (Fig. 2, Supplementary Figs. 1, 2).

***CTNNB1* mutations and confer phenotypic characteristics of DPN.** Some DPN arise juxtaposed with a common nevus[24–26]. We hypothesized genetic progression of a *BRAF*^V600E melanocyte within a common nevus by acquisition of a β-catenin activating mutation results in an adjacent DPN. In 66 out of 68 (97%) bi-phenotypic cases the DPN displayed increased nuclear β-catenin and cyclin D1 compared to the adjacent common nevus. β-catenin and cyclin D1 levels in the DPN remained uniform, whereas they decreased in the adjacent common nevus with distance from the epithelium (Fig. 3). We genotyped *BRAF*, *MAP2K1*, and *CTNNB1* in the DPN of 11 biphasic lesions. Activating *CTNNB1* mutations were identified in 10 out of 11 DPN within common nevi, usually in combination with *BRAF*^V600E (Supplementary Table 3). In 2 cases we genotyped the adjacent common nevus component and found identical *BRAF*^V600E mutations as in the adjacent DPN but no *CTNNB1* mutations. These results indicate that common nevi can evolve to DPN by acquiring β-catenin activating mutations.

We transduced immortalized mouse melanocytes (melan-a) with *BRAF*^V600E either alone or in combination with *CTNNB1*^S33F. Melanocytes expressing *BRAF*^V600E and *CTNNB1*^S33F were larger, more pigmented, and expressed higher levels of cyclin D1 than melanocytes expressing *BRAF*^V600E alone (Fig. 4), recapitulating the characteristic features of DPN. These experimental findings indicate that *CTNNB1* mutations are sufficient to induce the phenotypic changes characteristic of DPN in *BRAF* mutant melanocytes.

**DPN-like melanomas harbor additional oncogenic mutations.** Rare DPN metastasize with lethal outcomes[4, 27]. It is not known whether DPN that metastasize are genetically different from those that do not. We genotyped two metastases that originated from previously diagnosed DPN and identified that in addition to the characteristic combination of MAPK activating (*NRAS*^Q61R or *MAP2K1*^I103D/E203K) and *CTNNB1* mutation (Fig. 5, Supplementary Fig. 4, Supplementary Table 4) both harbored multiple DNA copy number alterations, a hallmark of melanoma[28]. One also harbored *TERT* promoter (C228T) and *TP53*^T86M mutations.

Some melanomas have features of DPN such as melanocytes with pigmented cytoplasm and numerous stromal melanophages. To determine whether DPN-like melanomas share genetic characteristics with DPN, we analyzed 6 cases. One melanoma harbored a *GNAQ*^R183Q activating mutation and truncating *BAP1* mutation with loss of heterozygosity and thus was re-classified as a blue nevus-like melanoma[29]. The five remaining melanomas all harbored MAPK pathway activating mutations in *BRAF* or *NRAS* and three had activating mutations in the β-catenin pathway (Fig. 5). In all cases, copy number alterations and additional oncogenic alterations were identified, including *TERT* promoter mutation or biallelic loss of *CDKN2A*.

**Discussion**

In summary, our study shows that DPN result from the combined mutational activation of the MAP kinase and β-catenin pathways. Some DPN evolve from pre-existing nevi by acquiring a β-catenin pathway mutation secondary to the initiating MAP kinase pathway mutation. Prior work demonstrates that β-catenin signaling

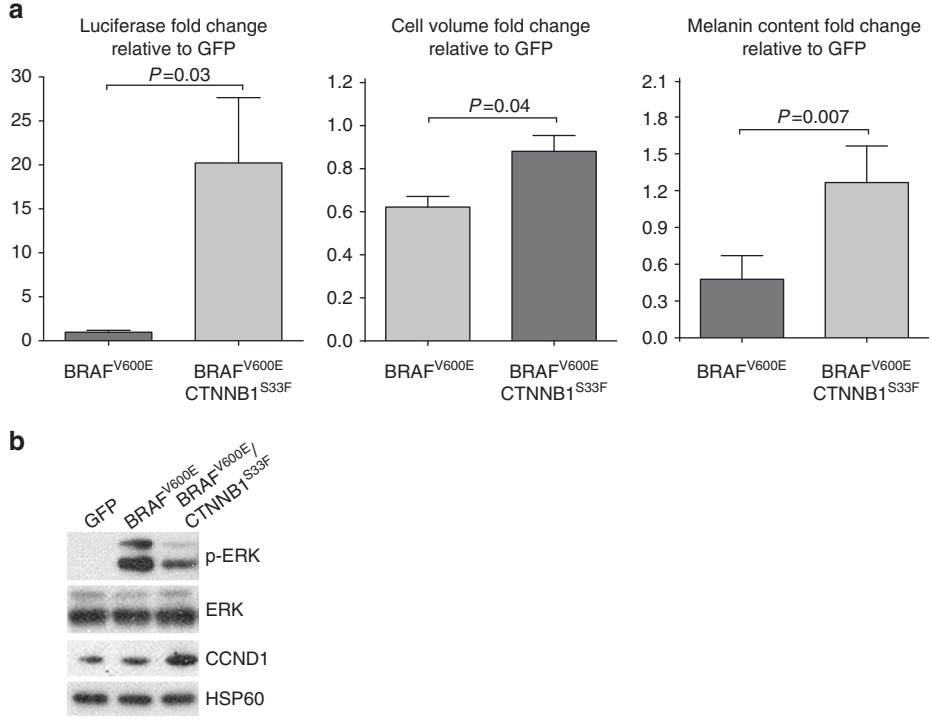

**Fig. 4** Mutant CTNNB1 confers increased cell volume, pigmentation and cyclin D1 levels in vitro. **a** Stably transduced melanocytes (melan-a) with BRAF[V600E] and CTNNB1[S33F] demonstrate increased β-catenin activity as measured by TOP flash luciferase expression (*left*; $P = 0.03$), cell volume (*center*; $P = 0.04$) and melanin content (*right*; $P = 0.007$) as compared to melanocytes transduced with BRAF[V600E] alone. Mean with *error bars* showing s.e.m., unpaired Student's *t*-test, three biological replicates performed. **b** Representative western blot analysis of melanocytes stably transduced with BRAF[V600E] and CTNNB1[S33F] demonstrate increased cyclin D1 levels as compared to those tranduced with GFP or BRAF[V600E]. Data shown are representative of at least three independent experiments with similar results

in melanocytes results in reduced T-cell response[30], which could contribute to the deeply invasive phenotype of DPN. The differences in β-catenin pathway activation between common nevi and DPN reveal another possible mechanism by which β-catenin pathway activation contributes to neoplastic progression. In DPN, melanocytes remain large and pigmented throughout the entire neoplasm, whereas in common nevi without mutational activation of the β-catenin pathway these cytomorphological features are constrained to areas in proximity to epithelia, and melanocytes become smaller and less pigmented with increasing distance from epithelia. Epithelia secrete WNTs[31], which may explain the decrease in β-catenin signaling with distance from epithelia. A dependence on WNT signaling may prevent common nevi from invading the deeper dermis. Constitutive β-catenin pathway activation in DPN may override this limitation, allowing these tumors to 'deeply penetrate'.

Our data further indicates that β-catenin and MAPK pathway mutations alone are insufficient to fully transform melanocytes, and that additional mutations such as immortalizing mutations of *TERT* and loss of *CDKN2A* are required for DPN to progress to melanoma, (Fig. 6) consistent with prior work in mouse models[32], [33]. Our results identify DPN as an intermediate melanocytic neoplasm, with a progression stage positioned between benign nevus and DPN-like melanoma.

## Methods

**Clinical Case Selection and Histopathologic Classification**. We searched the archives at the University of California Dermatopathology Unit and the personal consultation files of AdlF for cases annotated as having features of DPN. The study was approved by the Committee on Human Research of the University of California, San Francisco, with a waiver of patient consent. Consent to publish patient photos was obtained. The histopathology of all cases were reviewed by at least 2 expert dermatopathologists (AdlF, BCB, IY, ED) and categorized as either DPN,

nevus with overlapping features of DPN and blue nevus, 'combined nevi' with common nevus and DPN cytomorphology, or melanoma with features of DPN (markedly pigmented with defective maturation and occasionally deep invasion). Samples were excluded if the amount of tumor did not fall into one of the above predefined histopathologic subtypes or the tumor volume was deemed insufficient for further analysis. The number of samples in our cohort for sequencing was limited by the availability of DPN with enough residual material in our archives.

**DNA extraction and targeted next-generation sequencing**. Areas of tumor were microdissected from 20 μm sections of formalin-fixed, paraffin-embedded (FFPE) tumor. After deparaffinization by washing with Safeclear and ethanol DNA was extracted by phenol chloroform extraction. Multiplex library preparation was performed using the Ovation Ultralow Library System (NuGEN, San Carlos, CA, p/n 0331-32), Nextflex (Bioo Scientific, Austin, TX, p/n No. 5140-53) or Kapa Hyper Prep Kit (Kapa Biosystems, Wilmington, MA, p/n 07962363001) according to the manufacturer's specifications with up to 200 ng of sample DNA. Hybridization-capture of pooled libraries was performed using custom-designed bait libraries (Nimblegen SeqCap EZ Choice, p/n 06588786001) including the exons of *BRAF*, *NRAS*, *HRAS*, *KIT*, *GNAQ* and select introns of *BRAF*. Three versions of the custom-designed bait libraries were used, and the genes targeted and their overlap between versions are detailed in Supplementary Data 1–6. The target intervals cover mostly exonic but also some intronic and untranslated regions of 293 (version 1), 538 (version 2), and 480 (version 3) target genes. The target genes were curated to comprise common cancer genes with particular relevance to melanoma.

Captured libraries were sequenced as paired-end 100 bp reads on a HiSeq 2000 or HiSeq 2500 instrument (Illumina). Sequence reads were mapped to the reference human genome (hg19) using the Burrows-Wheeler aligner (BWA)[34]. Recalibration of reads was performed using the Genome Analysis Toolkit (GATK)[35]. Mutation calling was performed with FreeBayes[36]. Coverage and sequencing statistics were determined using Picard CalculateHsMetrics and Picard CollectInsertSizeMetrics[37] (Supplementary Table 5). Variant annotation was performed with Annovar[38] and variants with frequency in the 1000 genomes or exome sequencing project datasets of > 0.001 were excluded from further analysis. For fusion detection, read pairs with one or more reads unaligned, insert sizes greater than 1000 bp, or with soft clipping of at least one read were extracted and re-aligned using BWA-SW[39] and used as input to CREST[40]. Structural variants predicted by CREST were reviewed by visual inspection in the Integrative Genomics Viewer[41]. We predicted the resulting fusion transcript by joining the exon directly upstream from the genomic

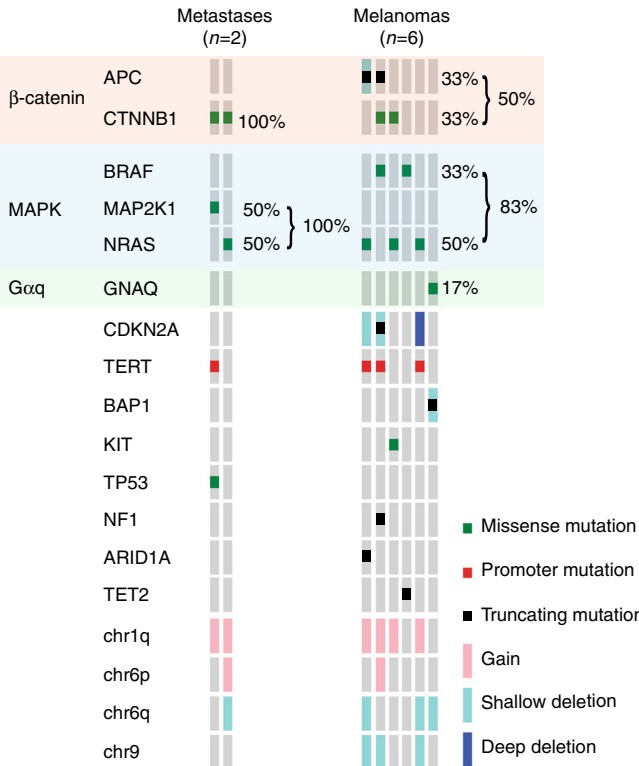

**Fig. 5** Additional genetic alterations in DPN-like melanoma. Distant metastases of DPN harbored a combination of MAPK and b-catenin pathway activating mutations in addition to oncogenic point mutations and copy number alterations, *left*. Half of primary melanomas with features of DPN harbored a combination of MAPK and b-catenin pathway activating mutations, also in combination with oncogenic point mutations and copy number alterations, *right*

breakpoint with the exon directly downstream. Predicted protein sequences were then determined from the predicted transcripts. Copy number analysis was performed using CNVkit[42].

**Statistical analysis**. We used the binomial test with the null hypothesis that β-catenin activating mutations are equally distributed between DPN and common nevi, and DPN and nevi with features of DPN and blue nevus. We also tested the null hypothesis that GNAQ mutations distinguish nevi with features of DPN and blue nevus from DPN.

**Plasmid construction**. Total RNA from 293FT cells (purchased from Life Technologies) was reverse transcribed with random primers, MAP2K1 cDNA was amplified with primer pairs 5'-caccatgcccaagaagaagccgac-2' and 5'-tta-gacgccagcagcatggg, and cloned into pENTR™/D-TOPO vector (ThermoFisher, p/n K240020). BRAF[V600E] and CTNNB1 cDNAs were cloned into pENTR™/D-TOPO using primer pairs 5'-caccatggcggcgctgagc and 5'-tcagtggacaggaaacgcacc-3', or 5'-caccatggctactcaagctgatttgatgg-3' and 5'-ttacaggtcagtatcaaaccaggc-3', respectively, and subsequently cloned into pLenti6.3/V5-DEST (i.e., BRAF, MAP2K1, and GFP constructs) or pLenti CMV/TO Puro DEST (670-1), a gift from Eric Campeau (Addgene plasmid #17293), (CTNNB1 construct)[43]. CTNNB1[S33F] and MAP2K1 mutations were introduced by site-directed mutagenesis using the following primer pairs: CTNNB1[S33F]: 5'-agcaacagtcttacctggactttggaatccattctg-3' and 5'-cagaatg-gattccaaagtccaggtaagactgttgct; MAP2K1[Q58_E62del], 5'-ctttcttacccagaagctga aggatgacgact-3' and 5'-agtcgtcatccttcagcttctgggtaagaaag; MAP2K1[E102_I103del], 5'-aaagctaattcatctgaaacccgcaatccgga-3' and 5'-tccggattgcgggtttcagatgaattagctttt-3'; MAP2K1[I103_K104del], 5'-aagctaattcatctgg aacccgcaatccggaa-3' and 5'-ttccggattgcg ggttccagatgaattagctt-3'; and MAP2K1[P105_I107delinsS], 5'-tcatctggagatcaaatccc ggaaccagatcata-3' and 5'-tatgatctggttccgggatttgatctccagatga-3'.

**Drug studies**. 293FT cells were transfected with MAP2K1 wild-type and mutant expression constructs using Jetprime (Polyplus-transfection, Illkirch) and 2 μg plasmid DNA as recommended by the manufacturer. Twenty four hours after transfection, the cells were treated for 4 h with 100 nM of trametinib (Selleckchem).

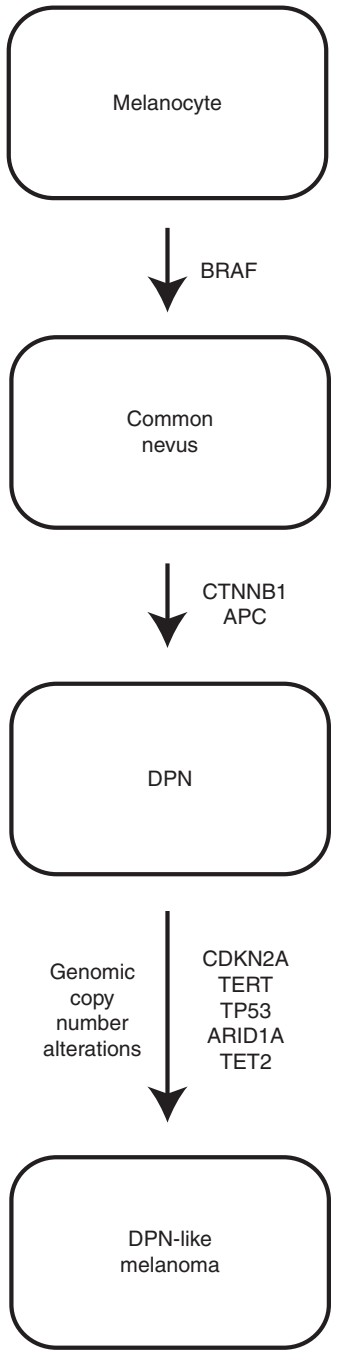

**Fig. 6** Model of step-wise progression in DPN-like melanoma. BRAF mutation leads to a common nevus. Subsequent CTNNB1 mutation results in the phenotypic switch to DPN. Additional genetic alterations result full transformation to DPN-like melanoma

**Immunohistochemistry**. Immunohistochemical analysis was performed on archival FFPE tumor specimens using the following antibodies: CTNNB1 clone B-catenin 1 (Dako catalog# M3539 or IR702), CCND1 clone EP-12 (Dako, catalog #IR083) or clone SP4 (Thermo Scientific, catalog #RM-9104-R7).

**RNA in situ hybridization**. Formalin-fixed, paraffin-embedded tissue sections were pretreated with 13 min of boiling followed by 12 min of protease digestion (RNAscope Protease Plus). Hybridization for hs-Axin2 (catalog #400249) was performed in parallel with DapB (catalog # 310043) and PPIB (catalog # 313901) as negative and positive controls respectively. Scoring for each case was performed in melanocytes near the epidermis and at a distance from epithelium (epidermis and adnexal epithelium). A semi-quantitative scoring method was used for each area (Supplementary Table 6)[44].

**Sanger sequencing**. Sanger sequencing was performed for *CTNNB1* exon 3, *BRAF* exon 15, *MAP2K1* exons 2 and 3. After selection of the relevant paraffin blocks or hematoxylin-eosin stained sections, genomic DNA was extracted from macro-dissected FFPE material (using the QIAamp DNA FFPE extraction kit (Qiagen, Courtaboeuf, France) according to the manufacturer's guidelines. Sanger sequencing analysis of BRAF exon 15, CTNNB1 exon 3, and MAP2K1 exons 2 and 3 were performed.

BRAF: Forward 5′-TCATAATGCTTGCTCTGATAGGA-3′; Reverse 5′-TCTA GTAACTCAGCAGCATCT-3′; CTNNB1: Forward 5′-TTAAAGTAACATTTCCA ATCTACTAATG-3′; Reverse 5′-TTCTTGAGTGAAGGACTGAGA-3′

MAP2K1, exon 2: Forward 5′-ACCTGAGCGTTTCTTTCCATGATA-3′ Reverse 5′-AATCAGTCTTCCTTCTACCCTG-3′

MAP2K1, exon 3: Forward 5′-CCTCTACCTTAAAGAGCTTAAACA-3′ Reverse 5′-TCACCTCCCAGACCAAAGATTA-3′.

**Generation of stably transduced cell lines**. Melan-a cells were generously provided by Dr Dorothy C. Bennett (St George's Hospital, University of London, London, UK)[45] and maintained in glutamine-containing RPMI-1640 supplemented with 10% heat-inactivated fetal bovine serum, 200 nM of 12-O-tetradecanoylphorbol-13-acetate (TPA), penicillin (100 units/ml) and streptomycin (50 mg/ml). 293FT cells were maintained in DME-H21 medium containing 10% heat-inactivated fetal bovine serum, MEM Non-Essential Amino Acids (0.1 mM), sodium pyruvate (1 mM), penicillin (100 units/ml) and strepto-mycin (50 mg/ml). Lentiviruses were produced by co-transfection of pLenti6.3/V5-DEST or pLenti CMV/TO Puro DEST (670-1) plasmid constructs with packaging vectors pCMV-VSV-G and pCMV delta R8.2 in 293FT cells. The supernatant containing viral particles were filtered on 0.45-µm polyvinylidene difluoride (PVDF) membranes and used to transduce melan-a cells in the presence of 10 µg/ml polybrene (Santa Cruz Biotechnology). Cells were selected using either 5 µg/ml blasticidin or 1 µg/ml puromycin for 1-3 weeks. Mycoplasma contamination was tested using MycoSensor PCR Assay Kit (Agilent Technologies).

**Western blotting**. Cells were lysed in RIPA buffer supplemented with Halt protease and phosphatase inhibitor cocktail (Thermo Scientific) and 5 mM EDTA, pH 8. Protein samples (20 µg) were electrophoresed in a NuPAGE 4–12% gradient Bis-Tris gel, transferred to a PVDF membrane. Antibodies were incubated in 5% milk-Tris-buffered saline, 0.1% tween-20 (TBST), washed in TBST, and bands were visualized by an enhanced chemiluminescent reagent and autoradiography. The antibodies used were: phospho-ERK (Cell Signaling, #9101; https://www.antibodypedia.com/gene/1203/MAPK3/antibody/167074/9101; http://1degreebio.org/reagents/product/862890/?qid=1704438), CTNNB1 (BD Biosciences, #610153; http://1degreebio.org/reagents/product/862646/?qid=1704449), BRAF (Santa Cruz Biotechnology, sc166; http://1degreebio.org/reagents/product/8775/?qid=1704450), MEK (Santa Cruz Biotechnology, sc436; http://1degreebio.org/reagents/product/8886/?qid=1704451), MEK (Santa Cruz Biotechnology, sc436; http://1degreebio.org/reagents/product/8886/?qid=1704451), phospho-MEK (Cell Signaling, #9121; https://www.antibodypedia.com/gene/3543/MAP2K2/antibody/107819/9121; http://1degreebio.org/reagents/product/809518/?qid=1704452), CCND1 (Santa Cruz Biotechnology, sc718; http://1degreebio.org/reagents/product/751352/?qid=1715632), and HSP60 (Santa Cruz Biotechnology, sc1722). All antibodies were used at 1:1000 dilution. Complete blots in Supplementary Fig. 5.

**TOPflash assay**. Stably transduced Melan-a cell lines cultured in 12-well plates were transfected with 1 ug of β-catenin reporter construct (M50 Super 8x TOP-Flash, a gift from Randall Moon, Addgene plasmid, #12456)[46] and renilla control reporter (pRL-CMV Vector, Promega, E2231). At 24 h after transfection, luciferase activity was assayed in 10 µl of lysate using the Dual-Luciferase reporter assay system (Promega, E1910) and a Glomax-Multi luminometer (Promega).

**Melanin quantitation**. Cells were trypsinized and centrifuged and the resulting cell pellet was dissolved in 1N NaOH-10% DMSO, and incubated at 74 °C for 2 h. The absorbance was measured at 420 nm. Melanin content was determined using a standard curve of melanin from *Sepia officinalis* (Sigma, #M2649).

**HoloMonitor imager measurements**. Stably transduced cell lines were plated in 6-well plates (Griener EK-27160) with 20,000 cells/well in serum free conditions. Digital holograms of the cells were generated 24–48 h after plating when they were at approximately 30% cell density using a HoloMonitor M4 Digital Holography Cytometer (Phase Holographic Imaging PHI AB, Lund, Sweden). Cell volume was calculated using Hstudio M4 software (Phase Holographic Imaging PHI AB, Lund, Sweden)[47].

**Data availability**. Sequence data that support the findings of this study have been deposited in the Sequence Read Archive (SRA), with the accession code BioProject ID: PRJNA384770 (http://www.ncbi.nlm.nih.gov/bioproject/384770). All other remaining data are available within the Article and Supplementary Files, or available from the authors upon request.

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

## Acknowledgements

We thank Richard Jordan for critical review of the manuscript and Tyrrell Nelson, Alexander Gagnon and Arthur DeLance for excellent technical assistance. We thank Dr Pauline Guyot for providing clinical and dermoscopic photographs. We thank Dr Andy Chien for generously providing reagents.

## Author contributions

I.Y., B.C.B. and A.d.l. F. designed the study, critically reviewed the data and wrote the manuscript. I.Y., A.H.S., V.H., D.P. and A.d.l.F. performed DNA sequence analysis. U.E.L., M.K.T., A.J., X.C. and R.L.J. performed in vitro studies. I.Y., B.C.B., A.d.l.F., E.D., L.C., P.E.L. and T.H.M. identified and reviewed the histopathological features of the study cohort. I.Y., A.d.l.F. and E.D. performed the data collection and analysis of immunohistochemical and in situ hybridization results.

## Additional information

**Competing interests:** The authors declare no competing financial interests.

