## [Peer Review File · Nature Communications]

Reviewers' comments:

Reviewer #1 (Remarks to the Author):

In this study the authors analysed 18 deep penetrating nevi for oncogenic genomic changes (single nucleotide variations, insertions/deletions, structural variants and copy number changes), using 10 common nevi, nevi with overlapping features of DPN and blue nevi, and melanomas with DPN-like features. They found that the majority of DPN harbored activating mutations in the beta-catenin pathway and the MAP-kinase pathway. The paper is well-written, and the supporting data are clear and convincing. The presence of MAP kinase pathway mutations in DPN have already been shown and is not novel (reference 6); however the involvement of the beta-catenin pathway is a novel finding, and the comparison of DPN to nevi with a DPN component as well as DPN-like melanoma is also interesting, detailed, and novel. This article will be of interest to those in both the clinical and research realms of melanocytic neoplasia.

The following corrections are recommended prior to publication:

- Line 18- Invade is not a good term to describe a benign nevus. Change deeply invade the skin, to `deeply extend through the dermis.
- Line 29 – The idea that DPN are intermediate between benign nevi and melanoma is too strong a conclusion based on a small case series. The majority of DPN do not progress to melanoma. Would soften this claim.
- Paragraph #2 (lines 47-49). Mention that DPN also have overlapping features with Spitz nevi and have been previously shown to harbor HRAS mutations in a subset of cases (reference 6).
- Line 65- Recommend elaborating further on how the presence of an IDH1 mutation may have changed the cytologic phenotype of a lesion that is genetically more similar to a Blue nevus.
- Figure 5 -Should include an H&E photo of the one of the DPN metastases and the original DPN it arose from, as metastasis in DPN is a rare occurrence and supporting substantiating data of the tumors cytology is necessary. Also genotyping of the original DPN from which the metastasis arose from would be helpful, if material is available.

Reviewer #2 (Remarks to the Author):

This is an outstanding manuscript describing the molecular characterization of deep penetrating nevi (DPNs), compared to benign nevi and melanoma. The investigators used targeted next generation sequencing that included a large number of genes (up to 538) known to carry mutations in melanomas. The major novel observations are 1) presence of activating exon 3 mutations in β -catenin (CTNNB1) in 94% of the cases, never reported before and 2) activating mutations in BRAF, MAP2K1 and HRAS in a mutually exclusive pattern. These molecular markers are important because DPN can be difficult to diagnose correctly and occasionally they can metastasize. The investigators indeed show that two metastatic DPNs that harbored additional mutations (TERT and TP53) and copy number alterations.

Suggestion:

The bands in Figures 1D and 4B can be hardly distinguished from the background. They can be improved by converting the Western blot images to grey scale (Figure 1) and increasing the brightness with Photoshop.

Reviewer #3 (Remarks to the Author):

Summary: The authors conduct a thorough investigation and characterization of deep penetrating nevus (DPN). They demonstrate a stepwise progression from common nevus to DPN and possibly melanoma, which can be explained through genetic alterations. DNA sequencing, cell line experiments, and immunohistochemistry was utilized to discover and support that a combination of MAPK pathway and CTNNB1 activating mutations are responsible for nevi to DPN progression. Additional alterations are required for DPN to progress into melanoma.

Impact: The authors present provocative data that can give insight into melanoma tumorigenesis. This data can be of great importance because MAPK inhibitors have already been shown to be effective in advanced melanoma but resistance has shown to be a challenging problem and a better understanding of the activating mutations may help overcome these obstacles.

Suggestions:

1. Sample size was low and possibly underpowered. It would be good to see a justification for the number of samples, and whether they have any statistical significance for sample size, i.e. some data has p values, others I did not see.
2. It would be interesting to see if ectopic WNT overexpression or ligand administration in common nevus (BRAF mutated) could cause progression from common nevus into DPN.
3. DPN-like melanoma characterization based not only on genetic but experimental testing in cell lines e.g. colony formation in soft agar, contact inhibition.

Overall: The authors adequately describe their findings of the stepwise progression of a melanocyte to DPN-like melanoma employing DNA sequencing, immunohistochemistry, and other lab techniques. The findings are strong, but see above for suggestions to make it absolutely convincing.

Reviewers' comments:

Reviewer #1 (Remarks to the Author):

In this study the authors analysed 18 deep penetrating nevi for oncogenic genomic changes (single nucleotide variations, insertions/deletions, structural variants and copy number changes), using 10 common nevi, nevi with overlapping features of DPN and blue nevi, and melanomas with DPN-like features. They found that the majority of DPN harbored activating mutations in the beta-catenin pathway and the MAP-kinase pathway. The paper is well-written, and the supporting data are clear and convincing. The presence of MAP kinase pathway mutations in DPN have already been shown and is not novel (reference 6); however the involvement of the beta-catenin pathway is a novel finding, and the comparison of DPN to nevi with a DPN component as well as DPN-like melanoma is also interesting, detailed, and novel. This article will be of interest to those in both the clinical and research realms of melanocytic neoplasia.

The following corrections are recommended prior to publication:

- Line 18- Invade is not a good term to describe a benign nevus. Change deeply invade the skin, to 'deeply extend through the dermis.

Thank you for this comment; we have made the suggested change.

- Line 29 – The idea that DPN are intermediate between benign nevi and melanoma is too strong a conclusion based on a small case series. The majority of DPN do not progress to melanoma. Would soften this claim.

We agree with the reviewer that the majority of DPN do not progress to melanoma and did not mean to imply that DPN are high-risk tumors. In other work our group has shown that genetically intermediate lesions exist on the spectrum between nevus and melanoma, and many of these are thought to be low risk tumors. Our claim that DPN are intermediate between nevi and melanoma comes from our finding of beta-catenin activating mutations in DPN and the known driver role of these mutations in melanoma and other cancers. We have changed the statement in the abstract to indicate that DPN are genetically intermediate (having additional oncogenic mutation as compared to nevi) and emphasize that DPN-like melanoma have additional genetic alterations.

- Paragraph #2 (lines 47-49). Mention that DPN also have overlapping features with Spitz nevi and have been previously shown to harbor HRAS mutations in a subset of cases (reference 6).

Thank you for pointing out this omission. We have added this as suggested.

- Line 65- Recommend elaborating further on how the presence of an IDH1 mutation may have changed the cytologic phenotype of a lesion that is genetically more similar to a Blue nevus.

We have added an additional statement to clarify our thinking on this point.

- Figure 5 -Should include an H&E photo of the one of the DPN metastases and the original DPN it arose from, as metastasis in DPN is a rare occurrence and supporting substantiating data of the tumors cytology is necessary. Also genotyping of the original DPN from which the metastasis arose from would be helpful, if material is available.

We agree that demonstration of the tumor cytology is necessary and have added a supplemental figure with details of one of the original DPN and its metastasis. We agree genotyping the original DPN from which the metastases occurred would be ideal. Unfortunately for both cases, material from the original DPN was exhausted. We cannot exclude the possibility that the original diagnoses of DPN were in error, although the histopathology was reviewed by multiple expert dermatopathologists. However, even if the primary tumors were misclassified as benign, the genetic findings from the metastases still contribute to our argument that genetic alterations in addition to MAPK and beta-catenin activating mutations are required for full transformation to DPN-like melanoma.

Reviewer #2 (Remarks to the Author):

This is an outstanding manuscript describing the molecular characterization of deep penetrating nevi (DPNs), compared to benign nevi and melanoma. The investigators used targeted next generation sequencing that included a large number of genes (up to 538) known to carry mutations in melanomas. The major novel observations are 1) presence of activating exon 3 mutations in β -catenin (CTNNB1) in 94% of the cases, never reported before and 2) activating mutations in BRAF, MAP2K1 and HRAS in a mutually exclusive pattern. These molecular markers are important because DPN can be difficult to diagnose correctly and occasionally they can metastasize. The investigators indeed show that two metastatic DPNs that harbored additional mutations (TERT and TP53) and copy number alterations.

Suggestion:

The bands in Figures 1D and 4B can be hardly distinguished from the background. They can be improved by converting the Western blot images to grey scale (Figure 1) and increasing the brightness with Photoshop.

Thank you for these suggestions, we have made the suggested changes.

Reviewer #3 (Remarks to the Author):

Summary: The authors conduct a thorough investigation and characterization of deep penetrating nevus (DPN). They demonstrate a stepwise progression from common nevus to DPN and possibly melanoma,

which can be explained through genetic alterations. DNA sequencing, cell line experiments, and immunohistochemistry was utilized to discover and support that a combination of MAPK pathway and CTNNB1 activating mutations are responsible for nevi to DPN progression. Additional alterations are required for DPN to progress into melanoma.

Impact: The authors present provocative data that can give insight into melanoma tumorigenesis. This data can be of great importance because MAPK inhibitors have already been shown to be effective in advanced melanoma but resistance has shown to be a challenging problem and a better understanding of the activating mutations may help overcome these obstacles.

Suggestions:

1. Sample size was low and possibly underpowered. It would be good to see a justification for the number of samples, and whether they have any statistical significance for sample size, i.e. some data has p values, others I did not see.

The cohort size of our study was limited by sample availability as we initially included tumors with only DPN cytomorphology and not “combined” nevi. However, despite the small cohort sizes, the observed distribution of beta-catenin activating mutations in DPN is highly statistically significant. We have added the statistics to the manuscript.

2. It would be interesting to see if ectopic WNT overexpression or ligand administration in common nevus (BRAF mutated) could cause progression from common nevus into DPN.

We agree that such experiment would confirm the phenotypic switch is directly related to WNT signaling. We currently lack animal models to perform this experiment but hope to collaborate with other teams to attempt to establish such models in the future.

3. DPN-like melanoma characterization based not only on genetic but experimental testing in cell lines e.g. colony formation in soft agar, contact inhibition.

We agree this would enhance our understanding of these tumors. The DPN-like melanoma tissues used in our study were all archival formalin fixed paraffin embedded diagnostic specimens, and no corresponding cell lines are available. We identified two melanoma cell lines in the literature with BRAF and CTNNB1 mutations (Yurif and SK-Mel-1). These cell lines were isolated from metastatic melanomas and have been characterized by other groups. We confirmed that these cell lines harbor additional genetic alterations as previously reported. We plan to use these cell lines to study the role of β -catenin signaling in fully transformed melanoma, but this is currently beyond the scope of our study demonstrating that CTNNB1 mutations occur as a secondary event in melanocytic neoplasia but do not lead to full transformation.

Overall: The authors adequately describe their findings of the stepwise progression of a melanocyte to DPN-like melanoma employing DNA sequencing, immunohistochemistry, and other lab techniques. The

findings are strong, but see above for suggestions to make it absolutely convincing.

Reviewers' Comments:

Reviewer #1 (Remarks to the Author):

Thank you for making the suggested changes. I recommend accepting this article for publication.

Reviewer #3 (Remarks to the Author):

The responses are sufficient.